# Insights into the Microbiome and Antibiotic Resistance Genes from Hospital Environmental Surfaces: A Prime Source of Antimicrobial Resistance

**DOI:** 10.3390/antibiotics13020127

**Published:** 2024-01-26

**Authors:** Alfizah Hanafiah, Asif Sukri, Hamidah Yusoff, Chia Sing Chan, Nur Hazlin Hazrin-Chong, Sharifah Azura Salleh, Hui-min Neoh

**Affiliations:** 1Department of Medical Microbiology and Immunology, Faculty of Medicine, Universiti Kebangsaan Malaysia, Cheras, Kuala Lumpur 56000, Malaysia; hamidah.yusoff@ppukm.ukm.edu.my; 2Department of Biological Sciences and Biotechnology, Faculty of Science and Technology, Universiti Kebangsaan Malaysia, Bangi 43600, Malaysia; hazlin@ukm.edu.my; 3ScienceVision Sdn Bhd, Shah Alam 40170, Malaysia; chiasing.chan@illumina.com; 4Infection Control Unit, Hospital Canselor Tuanku Muhriz, Cheras, Kuala Lumpur 56000, Malaysia; drazura@ppukm.ukm.edu.my; 5UKM Molecular Biology Institute (UMBI), Universiti Kebangsaan Malaysia, Cheras, Kuala Lumpur 56000, Malaysia; hui-min@ppukm.ukm.edu.my

**Keywords:** microbiome, antibiotic resistance genes, hospital environment, 16S rRNA, metagenome

## Abstract

Hospital environmental surfaces are potential reservoirs for transmitting hospital-associated pathogens. This study aimed to profile microbiomes and antibiotic resistance genes (ARGs) from hospital environmental surfaces using 16S rRNA amplicon and metagenomic sequencing at a tertiary teaching hospital in Malaysia. Samples were collected from patient sinks and healthcare staff counters at surgery and orthopaedic wards. The samples’ DNA were subjected to 16S rRNA amplicon and shotgun sequencing to identify bacterial taxonomic profiles, antibiotic resistance genes, and virulence factor pathways. The bacterial richness was more diverse in the samples collected from patient sinks than those collected from staff counters. Proteobacteria and Verrucomicrobia dominated at the phylum level, while *Bacillus*, *Staphylococcus*, *Pseudomonas*, and *Acinetobacter* dominated at the genus level. *Staphylococcus epidermidis* and *Staphylococcus aureus* were prevalent on sinks while *Bacillus cereus* dominated the counter samples. The highest counts of ARGs to beta-lactam were detected, followed by ARGs against fosfomycin and cephalosporin. We report the detection of *mcr-10.1* that confers resistance to colistin at a hospital setting in Malaysia. The virulence gene pathways that aid in antibiotic resistance gene transfer between bacteria were identified. Environmental surfaces serve as potential reservoirs for nosocomial infections and require mitigation strategies to control the spread of antibiotic resistance bacteria.

## 1. Introduction

Hospital-associated infections are a global public health problem that requires urgent intervention to mitigate the transmission of opportunistic pathogens to immunocompromised patients [1]. To address this issue, infection control measures that include disinfection of hospital environment surfaces, antibiotic stewardship, immunization of healthcare personnel, visitor management at hospitals, and hand washing have been implemented to control the transmission of hospital-associated pathogens [2]. These pathogens include *Staphylococcus aureus*, *Pseudomonas aeruginosa*, and *Acinetobacter baumannii*, which are able to form biofilms and resist killing by disinfectants. These pathogens’ resistance to antimicrobials allows them to survive on hospital surfaces, such as door handles, operating theatre floors, bathroom floors, and sinks, which abets the transmission of these pathogens in healthcare facilities [3]. Visitors and caretakers without adequate knowledge on infection control and the importance of hygienic practices in the hospital can also be a source of nosocomial infections by transmitting opportunistic pathogens from the community such as those from domestic animals, pets, and agriculture to healthcare facilities or vice versa [4,5]. While the COVID-19 pandemic has taught Malaysians about the importance of proper infection control to prevent the transmission of infectious diseases, the general public has become negligent post-pandemic [6]. Furthermore, the increase in antibiotic resistance rates against essential antibiotics has led to treatment failures in patients; the forward transmission of antibiotic-resistant bacteria from these patients to environmental surfaces in hospitals and other patients has been reported [7]. In 2019, the worldwide mortality associated with antibiotic resistance was 4.9 million, which mostly occurred in developing countries. It is estimated that infection control to reduce the transmission of antibiotic resistance bacteria can significantly decrease the mortality associated with this problem [8].

As part of infection control, surveillance of bacterial diversity and the presence of antibiotic resistance genes in the bacteria have been conducted using traditional methods that include culturing and amplification of targeted ARGs [9]. While these methods are useful in infection control to mitigate nosocomial infections, they are limited to specific bacterial groups and resistance genes. Hence, the actual representation of bacteria ARGs in the hospital is not adequately addressed [10]. In Malaysia, the current surveillance of bacterial diversity and antibiotic resistance genes in hospital environments mostly relies on polymerase chain reaction and the culture method. The advent of 16S rRNA amplicon and shotgun metagenomic sequencing aids in the surveillance process to mitigate nosocomial infections in healthcare facilities. These molecular methods are sensitive and reliable in the detection of resistance genes that have not been reported in specific hospital settings and unculturable bacteria that are currently present in the hospital settings [11]. The information obtained from this approach can also be used to identify the transmission source [12]. Using a metagenomics approach, bacterial cultures can also be pooled together to analyse the taxonomic profile of the bacteria as well as the comprehensive profile of antibiotic resistance genes of the bacteria in cost-effective manner. Here, we profiled the bacterial diversity and antibiotic resistance genes from hospital surface environments using these high-throughput, next-generation sequencing (NGS) approaches.

## 2. Results

### 2.1. Hospital Sample Collection

Hospital Canselor Tuanku Muhriz (HCTM) is a tertiary teaching hospital located in the urban area of Cheras, Kuala Lumpur. The hospital consists of 850 beds. In 2021, the hospital received 440,681 outpatients, whereas 29,874 patients were admitted to this hospital and 1425 patients underwent a surgical procedure(s) [13]. The samples from patient sinks (SP), staff sinks (SS), and staff counters (K) were collected from the Surgery and Orthopaedic wards. The results of the culture growth and Gram staining of the bacterial colonies are shown in Table 1. Gram-positive cocci were observed in the SP samples collected from wards 4G and 5F, while Gram-negative rods were identified in the SP samples collected from wards 4F, 5G, and 5B. On the other hand, Gram-negative diplococci was identified in 4FK.

DNA extraction was performed on all swab samples (n = 24); however, only 9 samples showed good quality and a high concentration of DNA that qualified for subsequent metagenomic and shotgun sequencing. These samples were from ward 4FSP, 4FK, 4GSP, 5FSP, 5FK, 5GSP, 5ESP, 5EK, and 5BSP.

### 2.2. Alpha and Beta Diversity of the Samples

A rarefaction analysis showed that the curves for most samples sequenced in this study increased and then flattened, indicating sufficient sequencing depths for the samples (Appendix A). The Shannon diversity index revealed that the sample collected at 4FSP had the highest diversity of bacteria while the lowest diversity was noted in the sample collected from 5BSP. A Chao1 analysis also demonstrated that sample 4FSP collected from patient sinks had the highest richness (Appendix A). The Jackknifed unweighted pair group method with arithmetic mean (UPGMA) analysis revealed that samples collected from 4FSP and 4FK were clustered together while samples collected in wards 5BSP and 5ESP belonged to another distantly related cluster (Appendix A). The bacterial diversity of the samples collected from 5FSP and 5FK were in a different cluster although they were collected from similar wards. The samples collected from 5EK and 5FK showed close relatedness to the sample collected from 4GSP.

### 2.3. Taxonomic Profiles at Phylum, Class, and Order Levels

At the phylum level, Firmicutes (51.2%) and Proteobacteria (48.7%) dominated the hospital surface samples in all wards examined. Proteobacteria constituted 99–100% of the bacterial population in the samples collected from 5BSP and 5ESP, while Firmicutes constituted 98–100% of the bacterial population from 5EK and 5FK and 4GSP (Figure 1A). In the class taxonomic ranking, Bacilli (51.2%) and Gammaproteobacteria (48.7%) dominated all samples collected in this study (Figure 1B). The Gammaproteobacteria abundance was prominent in the samples collected from patient sinks in the following wards: 4F (74.9%), 5B (99.8%), 5E (99.8%), and 5F (77.3%). The Bacilli abundance was prominent in samples collected from 5EK (99.8%) and 5FK (98.4%), and from 4GSP (100%) and 5GSP (56.7%). At the order level, Bacillales was the most abundant bacterial order detected (50.3%), followed by Pseudomonadales (31.9%), Enterobacteriales (7.4%), and Aeromonadales (6.0%) (Figure 1C). Bacillales dominated the samples collected from patient sinks in wards 4G and 5G, while Pseudomonadales dominated the samples collected from 5BSP and the counter surface in ward 4FK.

### 2.4. Taxonomic Profiles at Family, Genus, and Species Levels

The top five most dominant bacterial families detected in all samples included (1) *Staphylococcaceae* (33.3%), (2) *Pseudomonadaceae* (20.3%), (3) *Bacillaceae* (16.6%), (4) *Moraxellaceae* (11.6%), and (5) *Enterobacteriaceae* (7.4%) (Figure 2A). *Bacillaceae*, *Staphylococcaceae*, and *Moraxellaceae* dominated the bacterial abundance in the samples collected from counter surfaces in wards 4F, 5E, and 5F while *Pseudomonadaceae*, *Staphylococcaceae*, and *Xanthomonadaceae* were prominent in the patient sink samples collected from wards 5B, 4G, and 5F, respectively. At the genus level, *Staphylococcus*, *Pseudomonas*, *Bacillus*, and *Acinetobacter* were the top four most abundant bacteria detected in all samples. Notably, *Staphylococcus* was the most abundant (>98%) in the samples collected from a patient sink (ward 4G) and counter surface (ward 5F) (Figure 2B). *Acinetobacter* was the most abundant genus detected from the counter sample collected from ward 4F. At the species level, 72.9% of the reads belonged to unknown species. For known species, *Staphylococcus epidermidis* (11.4%), *Staphylococcus aureus* (4.4%), and *Bacillus cereus* (10.9%) were the top three most abundant species identified in all samples examined. *S. epidermidis* was dominant in patient sink samples collected from ward 4G and *B. cereus* was dominant in the counter surface sample collected from ward 5E (Figure 2C).

### 2.5. Antibiotic Resistance Gene Profiles

Appendix A show the sequencing outputs for the samples sequenced in this study for the identification of antibiotic resistance genes. The number of reads generated from the sequenced samples ranged from 27.2 to 32.3 million reads (minimum length of 150 bp) while the number of contigs generated (≥0 bp) ranged from 2380 to 76,931. The quality of sequencing reads that included N50, N75, L50, and L75 can be found in Appendix A. The relative abundance of antibiotic resistance genes from the different wards is shown in Figure 3.

The identification of antibiotic resistance genes conducted through de novo assembly using the AMRPlusFinder database revealed 43 different antibiotic resistance genes (Figure 4) that confer resistance to 20 distinct antibiotic classes. Most of the genes identified in this study shared >98% identity to the ones in the database with >90% coverage (Appendix A). Of the 43 antibiotic resistance genes detected, 29 genes were detected from sample 4FSP, and 11 genes from sample 4GSP and sample 4FK. The type of resistance with the highest number of antibiotic resistance genes was resistance to beta lactam antibiotics (n = 11), followed by fosfomycin (n = 7), and cephalosporin (n = 6) (Figure 4).

Further, we detected the *mcr10.1* gene, which confers resistance to colistin, which is the first time it was reported in a clinical setting in Malaysia. The gene encoding phosphoethanolamine-lipid A transferase MCR-10.1 was detected in the sample from 4FSP and is 100% identical to the gene encoded by *Enterobacter roggenkampii* (NCBI Reference Sequence: NG_066767.1). Furthermore, we found that *Enterobacter* sp. constituted 14.1% of the genus abundance in samples collected from 4FSP. However, our metagenomic analysis did not detect the species *Enterobacter roggenkampii* in the sample. For antibiotic resistance genes that confer resistance to beta lactam, we observed two variants of the *bla*_ACT_, *bla*_ADC_, *bla*_OXA_, and *blaR1* resistance genes (Figure 4). On the other hand, four variants of *fosA* conferring resistance to fosfomycin were discovered in the samples analysed. Two types of efflux genes that confer resistance to tetracycline, namely *tetC* and *tet38*, were detected.

### 2.6. Virulence Pathways from Hospital Environmental Samples

We interrogated the virulence pathways in the samples to identify the important pathways associated with resistance of hospital-associated pathogens to antibiotics using a KEGG pathway analysis. The lists of all virulence pathways detected in our study is available in Appendix A. We found that SP samples harboured more genes involved in virulence pathways than K samples (Figure 5). In total, the virulence pathways detected across all analysed samples included pathways important in the bacterial secretion system (n = 58), flagellar formation (n = 44), and biofilm formation (n = 5). The bacterial secretion system involved included type III (n = 22) and type IV (n = 13, and type VI (n = 23) secretion systems. Interestingly, the samples collected from 4FSP harboured a high abundance of virulence pathways that play a pertinent role in flagellar formation. This result was not observed in the K samples. Indeed, the samples collected from 5BSP harboured a high abundance of genes involved in pathways related to type III, IV, and type VI secretion system pathways (Figure 5), while the samples collected from 4FSP showed a high abundance of genes involved in pathways related to the type VI secretion system (secreted protein Hcp) and biofilm formation including biofilm poly-beta-1,6-N-acetyl-D-glucosamine N-deacetylase (PGA) synthesis N-glycosyltransferase PgaC, lipoprotein PgaB, and protein PgaD. Of note, sample 4FK exhibited a high abundance of one gene involved in biofilm formation, namely, biofilm PGA synthesis N-glycosyltransferase PgaC.

## 3. Discussion

Pathogenic bacteria from hospital environmental surfaces have the potential to be transmitted to patients, healthcare workers, and hospital visitors [14]. These bacteria may carry antibiotic resistance genes that further complicate treatment options, particularly for immunocompromised patients with indwelling medical devices [15]. Understanding the microbiome composition and its antibiotic resistance patterns from these environments facilitates infection control and prevention in healthcare settings [16]. In this study, we collected samples from patient sinks and staff counters because these sites potentially serve as a reservoir for the transmission of pathogenic bacteria to the patients and healthcare personnel. In addition, the orthopaedic and surgery wards were chosen because these wards record the highest infection rates in our hospital annually, and patients hospitalized there are immunocompromised with a high risk of mortality and morbidity. A previous study conducted in our hospital discovered that these wards were amongst the top three where MRSA isolates were isolated [17]. A previous study also showed that patients from surgery and orthopaedic wards in Malaysia usually did not comply with finishing the antibiotics prescribed to them, possibly contributing to the spread of antibiotic resistance [18]. In contrast to other studies from clinical settings in Malaysia that used PCR and culturing to identify bacteria and antibiotic resistance genes, we adopted 16S RNA amplicon and shotgun sequencing to provide insights into the diversity of the bacterial community and antibiotic resistance genes in our hospital. Our findings highlight that the bacterial community in our hospital that may be a source for nosocomial infections and antibiotic resistance; this includes resistance to colistin, which was reported here for the first time from a hospital environment in Malaysia. We chose to culture the swab first before DNA extraction in order to enrich the DNA for amplicon sequencing. We noticed that out of the 24 samples collected, only 9 samples resulted in growth of the bacteria with a sufficient DNA concentration for the amplicon sequencing. This is consistent with a previous study that found not all environmental swabs from hospital surfaces resulted in the growth of bacteria [19]. Regarding the concern that contamination might occur during sample processing, there are no currently available and clear guidelines to eliminate the contamination that might occur during sample collection for metagenomics studies. However, we attempted to minimize the contamination that might occur through (1) stringent aseptic technique during sample collection and transportation to our laboratory, (2) exclusion of samples with low DNA quantity that might be due to contamination, (3) inclusion of a negative control in our experiment, (4) incubation of swab culture for not more than 24 h, and (5) examination of negative and positive cultures using Gram staining to confirm the presence of bacteria. The contamination of samples from external sources, i.e., our laboratory, is unlikely because the culture of some swab samples resulted in no growth of bacteria.

The16S rRNA amplicon sequencing of samples from hospital environmental surfaces, particularly from sinks and countertops, demonstrated a low diversity and the samples were mostly dominated by hospital-associated pathogens that include *Staphylococcus*, *Pseudomonas*, *Acinetobacter*, and *Bacillus*. However, this finding should be interpreted with caution as we performed the analysis from the swab samples that were cultured in broth prior to DNA extraction to enrich the bacterial DNA for the sequencing. A further study to sequence the bacterial diversity directly from the swab should be conducted. Our findings on the diversity of bacteria at the genus level from our hospital environment were similar to the findings from a previous study [20]. A low diversity of the bacterial community from an indoor hospital environment compared to an outdoor hospital environment has been demonstrated in a previous study [21]. There are several factors that can influence the bacterial diversity in hospital environments including the use of antibiotics to treat infections, frequent disinfection and cleaning practices of hospital surfaces and equipment, and controlled temperature and humidity to suppress the growth of most bacteria [22]. However, these factors also contribute to the emergence and growth of hospital-associated pathogens that are resistant to antibiotics and disinfectants, and can survive unfavourable environments. Without proper infection control measures, such as good hygienic practice among healthcare workers and patients and regular disinfection of high-contact surfaces, surface environments including staff counters and patient sinks can serve as reservoirs for nosocomial infections. One limitation of 16S amplicon sequencing is that it cannot discriminate between some species of bacteria especially if those bacterial species share similar sequences. A previous study also showed that amplicon sequencing could not detect some bacterial species such as *Haemophilus influenzae*, but that species could be detected using a culture-based method [23]. It is imperative that 16S amplicon sequencing should be run in parallel with culturing methods in future studies to identify some bacterial species that cannot be discriminated using the sequencing approach.

We found that patient sinks from ward 4F harboured a more diverse bacterial community than that of staff counters. Patient sinks are rich in bacterial diversity because they are often used by patients and their visitors or caretakers to wash their hands, towels, and other equipment to clean patients. Furthermore, staff counters are always dry while patient sinks are always moist because of frequent use from patients and their visitors. Hence, they harbour a more diverse bacterial community compared to staff counters [24]. At the genus level, *Bacillus*, *Staphylococcus*, and *Acinetobacter* dominated both counter surfaced and patient sinks, while *Pseudomonas* only dominated the patient sinks. *Bacillus* has the ability to form spores to confer resistance to extreme environments such as high heat and disinfectants [25] while *Acinetobacter* [26] and *Staphylococcus* [27] have been shown to form biofilms, acquire microbicide resistance genes, and survive in unfavourable environments. *Pseudomonas* also demonstrates the ability to survive against disinfectant toxicity and in harsh environments through biofilm formation [28]. However, in this study, we found that *Pseudomonas* was dominant in patient sinks but not on counter surfaces, indicating that *Pseudomonas* was more susceptible to disinfectants than *Acinetobacter*, *Staphylococcus*, and *Bacillus*. A previous study revealed that *Pseudomonas aeruginosa* is the most susceptible to the disinfectants, namely alcohol-based and chlorohexidine-based disinfectants, and mixtures of both alcohol and chlorohexidine, compared to *S. aureus*, *Enterococcus faecalis*, and *Salmonella enteritidis* [29]. Although staff counters are always sanitized, they still harbour hospital-associated pathogens that can be life-threatening to immunocompromised patients. These pathogens can be transmitted to patients from staff contaminated with the pathogens via touching the counter surface. In our hospital, we usually used 70% ethanol to sanitize the hospital environmental surfaces. Thus, using a more potent disinfectant to disinfect the hospital environment surfaces than the one currently used in our hospital, as well as educating visitors and caretakers of patients on the importance of proper infection control and proper hygienic practice, are warranted.

Bacteria can take up naked DNA including antibiotic resistance genes from the environment and integrate it into their genome through horizontal gene transfer. We observed that antibiotic resistance genes that confer resistance to beta lactam were the most abundant genes found in our hospital environment, particularly from patient sinks. Beta lactam antibiotics include penicillin, cephalosporins, and monobactams, which are among the most widely used antibiotic classes to treat bacterial infections. A recent study conducted in our hospital showed that beta lactam antibiotics are the most prescribed antibiotics in our hospital, suggesting a contribution of this practice to the detection of the highest number of genes conferring resistance to beta lactam in our study [30]. These antibiotics’ antibacterial mechanism inhibits cell wall synthesis in Gram-negative and positive bacteria [31]. Distinct variants of *blaOXA* that confer resistance to beta lactam antibiotics were detected in our hospital environment. *blaOXA*-395 has been reported in multidrug-resistant *P. aeruginosa* [32] while *blaOXA*-699 was found in *A. baumannii* [33]. For cephalosporin resistance, *blaSHV*-27 that encodes extended beta lactamase in *K. pneumoniae* [34] was detected in patient sinks. Of note, *blaADC*-32 that confers resistance to cephalosporin in *A. baumannii* was detected on staff counters, indicating the need to properly disinfect counters to prevent the transmission of the resistant bacteria to patients. Only one carbapenem resistance gene, namely *BcII*, that confers resistance to carbapenem in *B. cereus* was detected in patient sinks. In Malaysia, *B. cereus* isolated from foods demonstrated resistance to trimethoprim, penicillin, and ampicillin [35] while a study conducted on carbapenem resistance is lacking. Nonetheless, *B. cereus* resistance to carbapenem isolated from foods in other countries is still low [36,37]. However, the detection of genes that confer resistance to carbapenem in *B. cereus* has the potential to lead to the emergence of carbapenem-resistant *B. cereus* in our hospital settings. Infection control measures, therefore, should be implemented to mitigate this problem.

Colistin is often used as a last resort to treat extensively drug-resistant bacteria [38]. The resistance of *P. aeruginosa* to colistin has been reported as significantly increasing in Malaysia, from 3.3% in 2020 to 8% in 2021 [39]. Studies on colistin resistance in other bacteria apart from *P. aeruginosa* in hospital settings in Malaysia are scarce. Nevertheless, the *mcr-1* gene that confers resistance to colistin has been detected in *E. coli* isolated from broilers in Malaysia [40]. In this study, we detected, for the first time, the presence of the mcr-10.1 gene in a hospital in Malaysia. Interestingly, the *mcr-10.1* gene observed in our study shares 100% similarity to that found in *E. roggenkampii*, suggesting the circulation of colistin resistance in bacteria apart from *P. aeruginosa*. The detection of *mcr-10.1* in our hospital environment suggests an urgent need to investigate the source organism from where this gene originated and for the enactment of environmental metagenomic surveillance in the hospital. Recently, the Malaysian government has agreed to ban the use of colistin in animal feeds to mitigate the emergence of colistin-resistant bacteria [41]. Since its first discovery in 2020 in an *E. roggenkampii* isolate [42], *mcr10* has been discovered in other bacteria that include *Klebsiella pneumoniae*, *Escherichia coli*, and *Enterobacter kobei*. *E. roggenkampii* harbouring *mcr10* has been shown to have a 4-fold increase in MIC to colistin. The resistance gene has been detected in humans, animals, and environmental sources. It is worth noting that this resistance gene has been detected in our neighbouring countries, namely Singapore and Vietnam [43]. The resistance to colistin poses public health challenge as it is the last resort used to treat *Enterobacteriaceae*. One limitation of our study is that the source and transmission of *mcr10* is unknown. However, it is possible that it might have been transmitted from community to our hospital environment, given the fact that it is prevalent amongst slaughterhouse workers. A future study should employ targeted culturing of bacteria that may harbour this resistance gene, followed by targeted gene amplification of this gene in those bacteria.

In this study, we identified virulence factors from the shotgun metagenomic data. We detected the presence of multiple factors are involved in flagella formation, secretion systems, and biofilm formation. The patient sinks harboured the highest abundance of virulence factors while staff counters harboured the least. Flagella are an important bacterial component for motility and adhesion to surfaces to aid in colonising indwelling medical devices [44]. Interestingly, three types of secretion systems, namely types III, IV, and VI, were discovered in this study. Type III, IV, and VI secretion systems are usually found in Gram-negative bacteria [45,46,47]. Secretion systems play a pertinent role in the transfer of antibiotic resistance genes in bacteria [48]. Biofilm formation is another crucial factor that leads bacterial resistance against antibiotics and disinfectants [49]. We found several pathways involved in biofilm formation that may play an important role in the bacteria’s resistance to antibiotics and disinfectants.

Since this study is the first pilot study conducted to determine the microbial diversity and antibiotic resistance genes of clinical concern in our hospital using amplicon sequencing and metagenomics, there are two limitations to this study. Firstly, the sample size in our study was small mainly because this is the first pilot study performed in our hospital for the surveillance of pathogenic bacteria and antibiotic resistance. However, our study serves as a pioneering study in employing metagenomics and amplicon sequencing in a hospital setting in Malaysia as part of a holistic “one health” approach for the surveillance of antibiotic resistance in our country to provide insight on intervention strategies to improve infection control. Secondly, while screening for the presence of antibiotic resistance genes in environmental surfaces is imperative to discover previously underreported or unreported resistance genes with clinical significance, it is also pertinent to determine which bacterial species harbour these resistance genes. This is because every bacterial species has different virulence factors and treatment guidelines for clinicians. However, another limitation of this study is that we did not perform an identification of the bacterial species. Future studies should identify the bacterial species that harbour the resistance genes of clinical concern through culturing and targeted sequencing of the resistance gene.

## 4. Materials and Methods

### 4.1. Sample Collection

According to Infection Control Unit of the Hospital Canselor Tuanku Muhriz, the annual data showed that hospital-associated infections occurred mostly in the Surgery and Orthopaedic wards. Hence, hospital environmental swab samples (n = 24) were collected from the Surgery [Surgery 1 (5B), Neuro (5E), Cardiothoracic (5G), and Ophthalmology (5F)] and Orthopaedic [Male Ortho (4F), Trauma Ortho (4G), Female Ortho (4H), and Spinal (4J)] wards. The samples were labelled according to the ward name, followed by the location where the samples were collected, where SP stands for patient sinks, SS stands for staff sinks, and K stands for staff counters. A duplicate of swabs from SP, SS, and K samples were collected from each ward. Swabs were firstly moistened with sterile normal saline, used to swab the sampling sites, and later immersed in sterile normal saline (0.9%). The swabs were vigorously vortexed and 10 μL of the sample suspension was transferred onto the culture media, namely blood agar, MacConkey agar, and mannitol salt agar, which are commonly used to isolate common hospital-associated pathogens. The plates were streaked and incubated at 37 °C overnight. Bacterial growth was identified by colony morphology and Gram staining. Another 10 μL of the sample suspension was inoculated into 1 mL of Brain Heart Infusion Broth (BHIB) and incubated at 37 °C overnight. After incubation, a drop of sterile glycerol was added and the culture was stored at −80 °C prior to DNA extraction. This procedure was conducted to enrich the DNA from the bacteria swabbed from the hospital environment [50].

DNA was then extracted using a commercial DNA kit (QIAGEN, Hilden, Germany) according to manufacturer’s instructions.

### 4.2. 16S rRNA Amplicon Sequencing

The extracted DNA was subjected to 16S rRNA metagenomic sequencing to identify the taxonomic profiles of the hospital environment swabs. Amplification of DNA was conducted using primers that targeted the 16S rRNA V3 and V4 hypervariable regions. The amplicons were subjected to metagenomic sequencing using the 16S Metagenomic Sequencing Library Preparation protocol (Illumina, San Diego, CA, USA). The amplicon library (550 bp) was generated using the SizeSelect^®^ 16S Amplicon Pre-Library gel system and barcoded using Illumina sequencing adapters from the Nextera XT Index kit (Illumina, USA). Subsequently, purification and library QC and quantification were performed using a Qubit Agilent Bioanalyzer and the KAPA Library Quantification Kit Illumina^®^ platforms. Library pooling and denaturation were carried out. Paired-end 16S rRNA metagenomics sequencing was carried out using the MiSeq Reagent kit v2 on the Illumina MiSeq NGS platform. FASTQ files (sequencing results) were generated using the MiSeq Reporter Software BaseSpace.

For the sequencing analysis, FASTQ files from BaseSpace were transferred into the BEPatho Software (BioEasy Sdn Bhd, Selangor, Malaysia). The BEPatho Software performed sequence alignment and the necessary nucleotide blasts to identify the bacterial genus and species. Output was generated in the form of bacterial abundance and identified bacterial genus and species using one-touch software. To determine the functions of the sequences including virulence pathways, PICRUSt2 was adopted [51] and mapped to the Kyoto Encyclopedia of Genes and Genomes (KEGG) pathways [52].

### 4.3. Shotgun Sequencing to Identify Antibiotic Resistance Genes

The extracted DNA from the hospital environmental swabs were subjected to shotgun sequencing to identify ARGs. Approximately 100 ng of gDNA (measured using a Qubit Fluorometer) was sheared to 350 bp using a Bioruptor. The sheared DNA was used as the input for Ultra II NEB Library Preparation (NEB, Ipswich, MA, USA). Briefly, the sheared DNA was end-repaired followed by Illumina adapter ligation and dual-index PCR to enrich for inserts with adapters at both ends. The library was quantified using qPCR and normalised for pair-end sequencing on a NovaSEQ6000 (Illumina, San Diego, CA, USA), generating on average 5 Gb of data for each sample.

### 4.4. Raw Read Processing

Raw sequencing reads were quality- and adapter-trimmed using Trimmomatic v.0.39 (default settings) [53].

### 4.5. De Novo Assembly and Identification of AMR Genes

The filtered reads were assembled de novo using Megahit (default option) [54]. The assembled contigs were assessed using QUAST [55] for the calculation of N50, GC%, and contig length distribution. Gene prediction was performed using prodigal v2.60 (-p meta) [56] followed by the identification of AMR-related genes using Abricate [57] against the NCBI AMRfinderplus database [58].

### 4.6. AMRplusplus2 Quantification of AMR Genes based on MEGARes Database

Using the AMRplusplus2 bioinformatic pipeline, the filtered reads were aligned against the MEGARes database [59] containing 8000 curated antimicrobial resistance genes, generating a count table containing the number of reads mapped to each AMR gene which can be used to calculate its relative abundance for the estimation of AMR prevalence in the sequenced samples.

## 5. Conclusions

In conclusion, our study demonstrates the urgency for intervention to mitigate antibiotic resistance in our hospital. We report the alarming data on the occurrence of antibiotic resistance genes including *mcr10.1* in the hospital environment, specifically at a sink used by hospital patients. This infers that visitor and patient caretaker awareness of antibiotic resistance is important to prevent the transmission of pathogenic bacteria in the hospital. Measures to identify the source of the resistance genes from hospital environments that include sinks and counter surfaces should be carried out for infection control purposes. Furthermore, surveillance of the incidence of antibiotic resistance bacteria, particularly resistance to essential antibiotics and their resistance mechanism, should be continuously monitored.

## Figures and Tables

**Figure 1 antibiotics-13-00127-f001:**
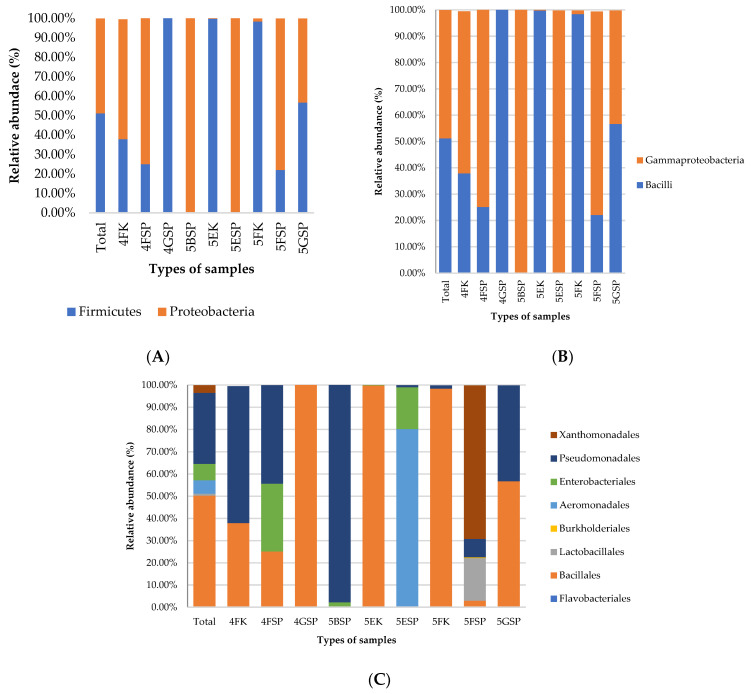
Distribution of bacterial community collected from the patient’s sinks and staff counters at (**A**) phylum, (**B**) class, and (**C**) order levels.

**Figure 2 antibiotics-13-00127-f002:**
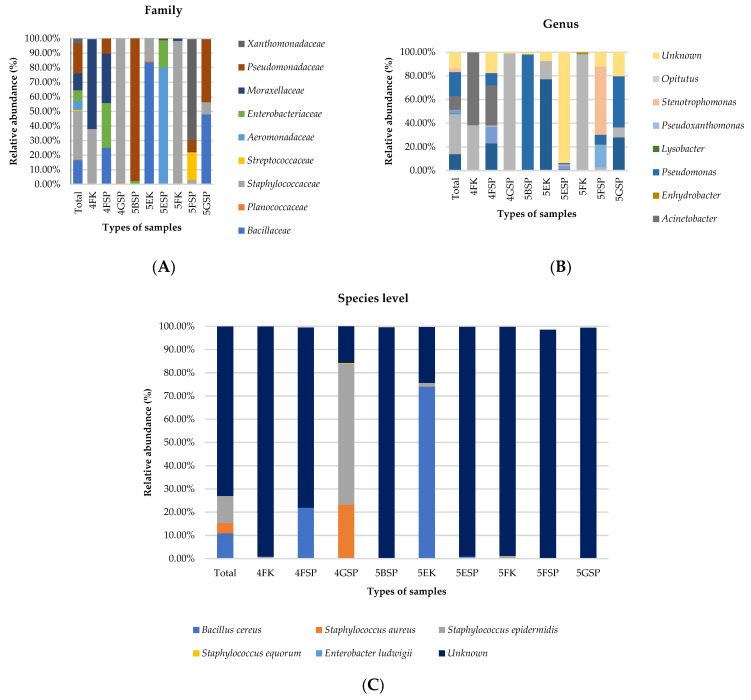
Distribution of bacterial community at (**A**) family, (**B**) genus, and (**C**) species levels in the samples collected from patient sinks and staff counters.

**Figure 3 antibiotics-13-00127-f003:**
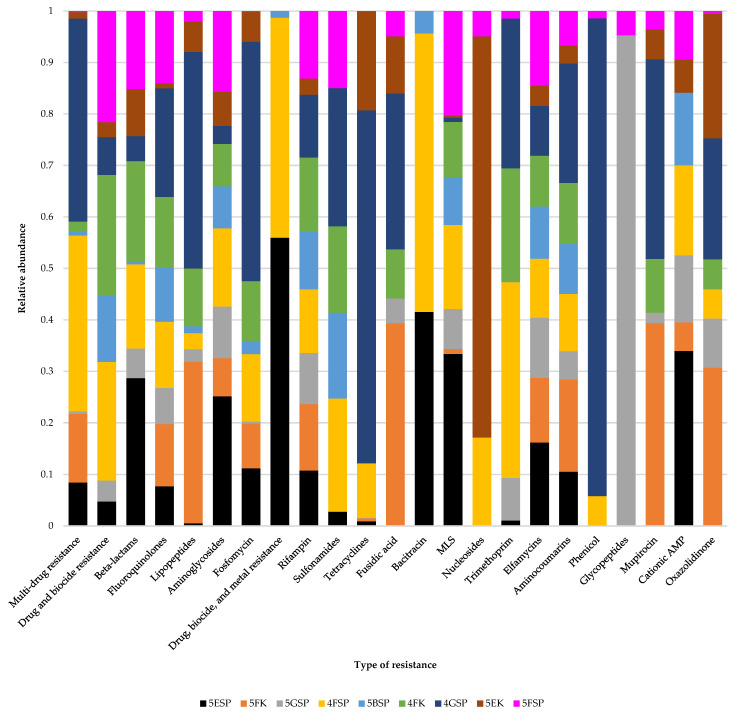
Relative abundance of antibiotic resistance genes identified using de novo assembly.

**Figure 4 antibiotics-13-00127-f004:**
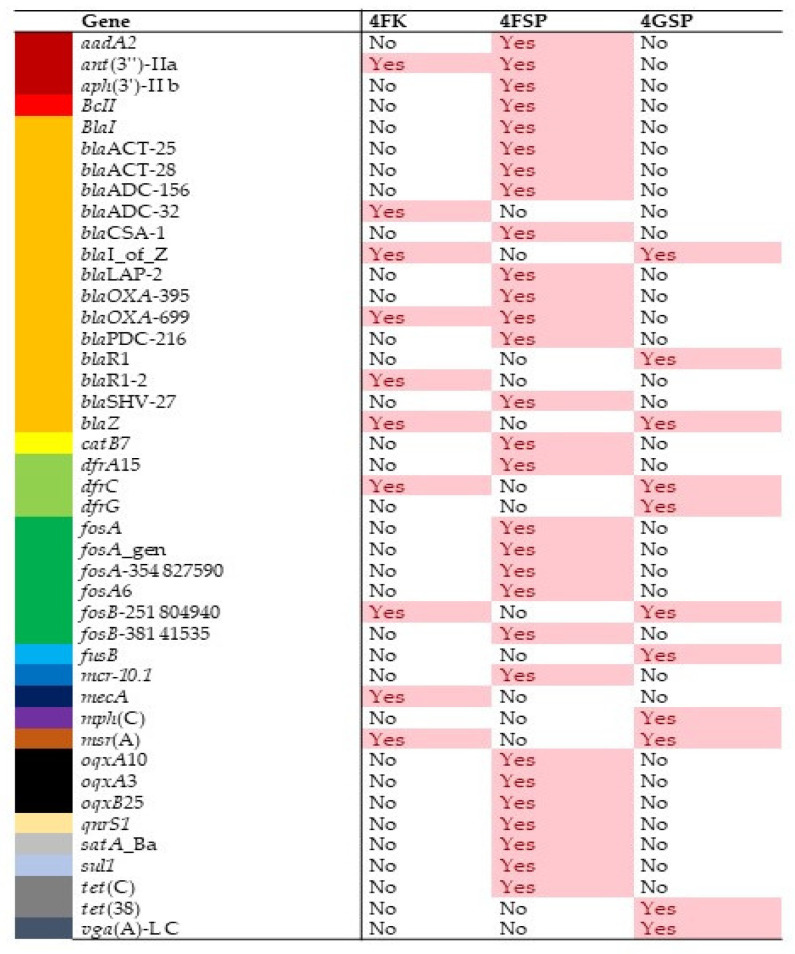
List of antibiotic resistance genes detected from shotgun metagenomic sequencing of the samples collected from patient sinks and staff counters. The list of resistance genes detected are colour-coded by antibiotic resistance class. “Yes” indicates that the gene was detected while “No” indicates that the gene was not detected in the sample examined. 4FK: sample collected from staff counter in ward 4F; 4FSP: sample collected from patient sink in ward 4F; and 4GSP: sample collected from patient sink in ward 4G.

**Figure 5 antibiotics-13-00127-f005:**
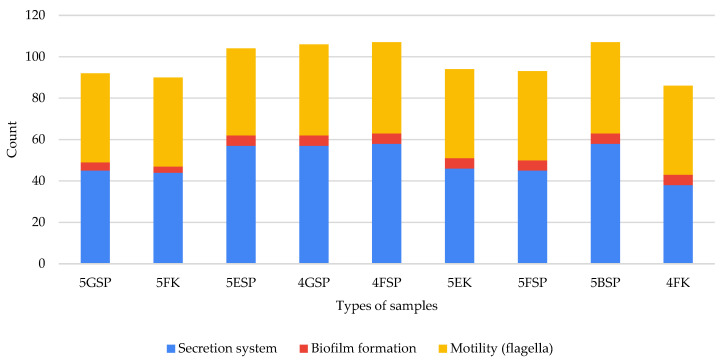
Number of pathways, including those related to secretion systems, biofilm formation, and motility, that virulence factors detected in respective samples are involved in.

**Table 1 antibiotics-13-00127-t001:** Types of samples and results of Gram staining of bacterial colonies isolated from respective samples. SP: patient sink; SS: staff sink; K: counter; GNR: Gram-negative rods; GNC: Gram-negative cocci; GPC: Gram-positive cocci; GNC: Gram-negative cocci; GN Diplo: Gram-negative diplococci; NG: no growth.

No.	Ward	Location	Blood Agar	MacConkey Agar	Mannitol Salt Agar
1.	4F	SP	GNR	GNR	GNR in chains
		SS	NG	GPR	NG
		K	Mixed growth positive/negative cocci	GN Diplo	NG
2.	4G	SP	GPC in cluster	NG	Mixed growth positive/negative cocci
		SS	NG	GNR	NG
		K	NG	NG	NG
3.	4H	SP	NG	NG	NG
		SS	NG	NG	NG
		K	NG	NG	NG
4.	4J	SP	NG	NG	NG
		SS	NG	NG	NG
		K	NG	NG	NG
5.	5F	SP	GPC/GNR	GNR	NG
		SS	GNC in pairs	NG	NG
		K	GPC/GNC	GPR	GPC
6.	5G	SP	Growth	GNR	GNR
		SS	NG	NG	NG
		K	NG	NG	NG
7.	5E	SP	GNC	GNC	GNC
		SS	NG	NG	NG
		K	GPC	NG	GPC
8.	5B	SP	GNR	GNR	mixed growth
		SS	NG	NG	NG
		K	NG	NG	NG

## Data Availability

Data are contained within the article.

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
