# Peer review of "Insights into the Microbiome and Antibiotic Resistance Genes from Hospital Environmental Surfaces: A Prime Source of Antimicrobial Resistance"

_antibiotics, 2024, doi:10.3390/antibiotics13020127_

Round 1
Reviewer 1 Report
Comments and Suggestions for Authors
The research covers a study on the microbial community and antibiotic resistance genes in a hospital environment. The research reveals that ward environments, particularly patients' sinks and counter surfaces, could serve as potential sources for the spread of pathogens, with the detection of various antibiotic resistance genes. Techniques such as 16S rRNA and Shotgun sequencing were employed to unveil the diversity of microbial communities and the distribution of antibiotic resistance genes across different wards. This paper is very substantial in content and uses a variety of methods. However, the presentation should be further improved before it can be published. To help authors improve the quality of the manuscript, the detailed suggestions are as follows:
some major concerns:
1. The title of the article is: “Insights into the microbiome and antibiotic resistance genes from hospital environmental surfaces: A prime source of antimicrobial resistance”. A key point in the results is that the mcr-10.1 gene was detected in the patient's sink, which was similar to that of E. roggenkampii, but no details are provided about the source, the route of transmission, or the potential public health impact. How important is this finding for the prevalence and spread of antibiotic resistance? Otherwise, the innovation of this study is still conservative.
2. The research mentions the choice of Surgery and Orthopedic wards as study areas. Are there specific data to support the high risk of infection in these areas? It’s better to add references or as additional data.
3. The study mentioned the monitoring of bacterial diversity and antibiotic resistance genes in the hospital setting and showed that beta-lactam antibiotic resistance genes were most abundant. Has this result been compared with similar studies already, or is it a phenomenon unique to the region? Does this mean that beta-lactam antibiotics are widely used in the hospital setting, resulting in the spread of resistance genes?
Some special comments:
1. Line 31. The authors should change “beta lactam” to “beta-lactam”. And check and revise the full text.
2. Line 40. The authors should change “require” to “requires”.
3. Line 56, etc. It should be like “post-pandemic” rather than “post-pandemic”.
4. Line 58. The authors should change “antibiotic resistant bacteria” to “antibiotic-resistant bacteria”.
5. Line 79. The authors should change “high-through put” to “high through-put”.
6. Line 189. “List of all virulence pathways”. Please note the plural of “List” here.
7. Line 251. There should be “they harbour a more diverse bacterial community”.
8. Line 287. “while study conducted on carbapenem” should change into “while the study conducted on carbapenem”. Otherwise, the predicate verb should be plural: “while studies conducted on carbapenem resistance are lacking”.
9. Line 282. “A. baumannii was detected on staff counters, indicating the need to properly disinfect counters to prevent transmission of the resistant bacteria to patients”. But according to the previous discussion, staff counters are considered to be a place that is often disinfected compared to Patients' sinks?
10. The antibiotic resistance genes collected from patient's sink and staff's counter are listed in Figure 4, where the genes should be italicized.
Author Response
Reviewer 1
The research covers a study on the microbial community and antibiotic resistance genes in a hospital environment. The research reveals that ward environments, particularly patients' sinks and counter surfaces, could serve as potential sources for the spread of pathogens, with the detection of various antibiotic resistance genes. Techniques such as 16S rRNA and Shotgun sequencing were employed to unveil the diversity of microbial communities and the distribution of antibiotic resistance genes across different wards. This paper is very substantial in content and uses a variety of methods. However, the presentation should be further improved before it can be published. To help authors improve the quality of the manuscript, the detailed suggestions are as follows:
some major concerns:
- The title of the article is: “Insights into the microbiome and antibiotic resistance genes from hospital environmental surfaces: A prime source of antimicrobial resistance”. A key point in the results is that the mcr-10.1gene was detected in the patient's sink, which was similar to that of roggenkampii, but no details are provided about the source, the route of transmission, or the potential public health impact. How important is this finding for the prevalence and spread of antibiotic resistance? Otherwise, the innovation of this study is still conservative.
Thank you for this comment. Since its first discovery in 2020 in E. roggenkampii isolate, mcr10 has been discovered in other bacteria that include Klebsiella pneumoniae, Escherichia coli, and Enterobacter kobei. E. roggenkampii harboring mcr10 has been shown to have 4-fold increase in MIC to colisin. The resistance gene has been detected in humans, animals, and environmental sources. It is worth noting that this resistance gene has been detected in our neighboring countries, namely Singapore and Vietnam. Resistance to colistin poses public health challenge as it is the last resort used to treat Enterobacteriaceae. One limitation of our study is the source and transmission of mcr10 in our hospital was unknown. However, it is possible that it might be transmitted from community to our hospital environment, given the fact that it is prevalent amongst slaughterhouse workers. Future study should employ targeted culture on bacteria that may harbour this resistance gene, followed by targeted gene amplification of this gene in those bacteria. We included the discussion on this limitation in our study from line 306-318.
- The research mentions the choice of Surgery and Orthopedic wards as study areas. Are there specific data to support the high risk of infection in these areas? It’s better to add references or as additional data.
Thank you for this suggestion. Apart from unpublished data where surgical and orthopaedic wards reported the highest infection rate annually in our hospital, we collected the samples from Surgery and Orthopaedic wards because previous study conducted in our hospital discovered that those wards were amongst the top three where MRSA isolates were isolated. Previous study conducted in the largest hospital in Kuala Lumpur, Malaysia also showed that patients from Surgery and Orthopaedic wards usually did not finish consuming antibiotics prescribed to them, possibly contributing to the spread of antibiotic resistance. We revised the manuscript to include those statements in Discussion section from line 221-225 .
- The study mentioned the monitoring of bacterial diversity and antibiotic resistance genes in the hospital setting and showed that beta-lactam antibiotic resistance genes were most abundant. Has this result been compared with similar studies already, or is it a phenomenon unique to the region? Does this mean that beta-lactam antibiotics are widely used in the hospital setting, resulting in the spread of resistance genes?
Thank you for this comment. Recent study conducted in our hospital showed that beta lactam antibiotics are the most prescribed antibiotics in our hospital, suggesting the contribution of this practice to the highest detection of genes conferring resistance to beta lactam in our study. We included this explanation in Discussion section from line 282-285 of our revised manuscript.
Some special comments:
- Line 31. The authors should change “beta lactam” to “beta-lactam”. And check and revise the full text.
We agree and revised as suggested.
- Line 40. The authors should change “require” to “requires”.
We agree and revised as suggested.
- Line 56, etc. It should be like “post-pandemic” rather than “post-pandemic”.
We agree and revised as suggested.
- Line 58. The authors should change “antibiotic resistant bacteria” to “antibiotic-resistant bacteria”.
We agree and revised as suggested.
- Line 79. The authors should change “high-through put” to “high through-put”.
We agree and revised as suggested.
- Line 189. “List of all virulence pathways”. Please note the plural of “List” here.
We agree and revised as suggested.
- Line 251. There should be “they harbour a more diverse bacterial community”.
We agree and revised as suggested.
- Line 287. “while study conducted on carbapenem” should change into “while the study conducted on carbapenem”. Otherwise, the predicate verb should be plural: “while studies conducted on carbapenem resistance are lacking”.
We agree and revised as suggested.
- Line 282. “A. baumanniiwas detected on staff counters, indicating the need to properly disinfect counters to prevent transmission of the resistant bacteria to patients”. But according to the previous discussion, staff counters are considered to be a place that is often disinfected compared to Patients' sinks?
Thank you for this comment. We apologize for the use of incorrect term. The term should have been “sanitize” which is the process to reduce the number of bacteria on surface. Therefore, we change the term “disinfect” to “sanitize” at Line 264. In our hospital, we often use 70% ethanol to sanitize the hospital environmental surfaces that include staff’s counter. We added this information at line 267-268. Although staff counter is often properly sanitized, Acinetobacter baumannii can still survive sanitization using 70% ethanol through biofilm formation. Thus, it is important in infection control measure to review the process of sanitization using different disinfectants and antiseptics especially when it involves bacteria that can form biofilm to survive harsh environmental conditions.
- The antibiotic resistance genes collected from patient's sink and staff's counter are listed in Figure 4, where the genes should be italicized.
Thank you for this suggestion. We agree with this suggestion and revised Figure 4 as suggested.
Reviewer 2 Report
Comments and Suggestions for Authors
The manuscript is to insight into the microbiome and antibiotic resistance genes from hospital enviromental surfaces.However,only 24 samples were collected from 8 wards,and DNA extraction only from 9 samples were qualified for meta-genomic and shotgun seqencing,the numbers of samples collected from hospital were so less ,which can not support the conclusion.The authors should enlarge the sample numbers.
Comments on the Quality of English LanguageThe English language of this manuscript is smoothly.
Author Response
Reviewer 2
The manuscript is to insight into the microbiome and antibiotic resistance genes from hospital enviromental surfaces.However,only 24 samples were collected from 8 wards,and DNA extraction only from 9 samples were qualified for meta-genomic and shotgun seqencing,the numbers of samples collected from hospital were so less ,which can not support the conclusion.The authors should enlarge the sample numbers.
Thank you for this comment. We admit that sample size in our study was small mainly because this is the first pilot study performed in our hospital for surveillance of pathogenic bacteria and antibiotic resistance in our hospital. However, our study serves as a pioneering study to employ metagenomics and amplicon sequencing in hospital setting in Malaysia as part of holistic “one health” approach for surveillance of antibiotic resistance in our country to provide insight on intervention strategy to improve infection control. Thus, we revised the manuscript to include small sample size limitation of our study in Discussion section (line 362-270).
Reviewer 3 Report
Comments and Suggestions for Authors
Hanafiah et al. present in their manuscript data towards an important topic - detection and surveillance of antibiotic resistances in a hospital setting. However, the presentation of results is lacking some important information. Mainly, the reasoning for culturing swabs. Usually, one would assume that if a contamination of microbial species is taking place a swap would result in enough material to sequence directly from the native sample. The authors, however, fail to show data why they chose to culture. Secondly, the results described show a lack of method validation. It is by now known that especially shotgun metagenomics can suffer from contamination (as does 16s amplicon sequencing). Thus, the results need to be cleaned and surveilled for contamination brought in either by laboratory consumables (this can be kit as well as culture contaminants) or contamination from the hospital, i.e. swaps. Most importantly, sterile tools do not mean DNA free tools. The authors describe that 29 samples were taken, however, only 8 have been analyzed due too poor quality. Such a high amount of drop out samples indicate a problem in sample acquisition. This circumstance is, hwoever, not addressed.
Figure 2C showing mostly unknown species does not contribute to the overall soundness of the paper. What is the message the authors want to verbalize by showing this?
It is well known, that by horizontal gene transfer resistances can hop from one species/ genus to another. However, the authors wonder why they do no see the appropriate (i.e. database assigned) species when performing AMR analysis. If this is a problem to the authors they should discuss in detail why. For example, is the species which harbours the resistance more important the the detection of the resistance itself - especially in a clinical context.
Given that the study was conducted in cooperation with the Infection control unit of the hospital it would be really great to see if there are any implication made towards hospital hygiene etc. based on the results of this study.
Comments on the Quality of English LanguageSome minor revision (typos) is necessary to improve readability.
Author Response
Reviewer 3
Hanafiah et al. present in their manuscript data towards an important topic - detection and surveillance of antibiotic resistances in a hospital setting. However, the presentation of results is lacking some important information. Mainly, the reasoning for culturing swabs. Usually, one would assume that if a contamination of microbial species is taking place a swap would result in enough material to sequence directly from the native sample. The authors, however, fail to show data why they chose to culture. Secondly, the results described show a lack of method validation. It is by now known that especially shotgun metagenomics can suffer from contamination (as does 16s amplicon sequencing). Thus, the results need to be cleaned and surveilled for contamination brought in either by laboratory consumables (this can be kit as well as culture contaminants) or contamination from the hospital, i.e. swaps. Most importantly, sterile tools do not mean DNA free tools. The authors describe that 29 samples were taken, however, only 8 have been analyzed due too poor quality. Such a high amount of drop out samples indicate a problem in sample acquisition. This circumstance is, hwoever, not addressed.
Thank you for this comment. We chose to culture the swab first before DNA extraction in order to enrich DNA for amplicon sequencing. We noticed that out of 24 samples collected, only 8 samples resulted in growth of bacteria with sufficient DNA concentration for the amplicon sequencing. This is consistent with previous study that found not all environmental swabs from hospital surfaces resulted in growth of bacteria. Regarding the concern that contamination might occur during sample processing, there is no clear guideline available currently to eliminate contamination that might occur during sample collection for metagenomics study. However, we attempted to minimize the contamination that might occur through 1) stringent aseptic technique during sample collection and transportation to our laboratory, 2) exclusion of samples with low DNA quantity that might be because of contamination, 3) inclusion of negative control in our experiment, 4) incubation of swab culture for not more than 24 hours and 4) examination of negative and positive culture using Gram stain to examine for the presence of bacteria. Contamination of samples from external sources i.e., our laboratory might be unlikely because some swab samples resulted in no growth of bacteria. We included the discussion of this comment from the reviewer in Discussion section (line 231-245) in our revised manuscript.
Figure 2C showing mostly unknown species does not contribute to the overall soundness of the paper. What is the message the authors want to verbalize by showing this?
Thank you for this comment. One limitation of 16S amplicon sequencing is it cannot discriminate some species of bacteria especially if those bacterial species share similar sequences at 16S region. Previous study also showed that amplicon sequencing could not detect some bacterial species such as Haemophilus influenzae, but that species could be detected using culture-based method. Thus, we want to highlight that it is imperative that 16S amplicon sequencing should be run in parallel with culturomics in the future study to identify some bacterial species that cannot be discriminated using sequencing approach. We included this matter in Discussion section of our revised manuscript (line 264-271).
It is well known, that by horizontal gene transfer resistances can hop from one species/ genus to another. However, the authors wonder why they do no see the appropriate (i.e. database assigned) species when performing AMR analysis. If this is a problem to the authors they should discuss in detail why. For example, is the species which harbours the resistance more important the the detection of the resistance itself - especially in a clinical context.
Thank you for this comment. While screening the presence of antibiotic resistance genes in environmental surfaces is imperative to discover previously underreported or unreported resistance genes with clinical significance, it is also pertinent to find out which bacterial species harbors those resistance genes. This is because every bacteria species has different virulence factors and treatment guidelines by clinicians to eradicate the bacteria from patients. However, one limitation of this study is we did not perform the identification of bacterial species. Future study should identify bacterial species that harbors resistance genes with clinical concern through culturomics and targeted sequencing of the resistance gene. We included the discussion on this matter in Discussion section of our revised manuscript (line 370-377).
Given that the study was conducted in cooperation with the Infection control unit of the hospital it would be really great to see if there are any implication made towards hospital hygiene etc. based on the results of this study.
Thank you so much for this comment.
Round 2
Reviewer 1 Report
Comments and Suggestions for Authors
The authors have answered the reviewer's previous questions and made appropriate revisions to the article. The current version is acceptable.
Reviewer 3 Report
Comments and Suggestions for Authors
The authors addresses all revision points appropriately.